# Profile, Infection, and Vaccination Uptake: A Cohort of Canadian Retail Workers During the SARS-CoV-2 Pandemic

**DOI:** 10.3390/idr17050122

**Published:** 2025-09-29

**Authors:** Mathieu Thériault, Kim Santerre, Nicholas Brousseau, Samuel Rochette, Rabeea F. Omar, Joelle N. Pelletier, Caroline Gilbert, Jean-François Masson, Mariana Baz, Denis Boudreau, Sylvie Trottier

**Affiliations:** 1Centre de Recherche en Infectiologie, Université Laval, Québec, QC G1V 4G2, Canada; kim.santerre.cemtl@ssss.gouv.qc.ca (K.S.); rabeea.omar@crchudequebec.ulaval.ca (R.F.O.); caroline.gilbert@crchudequebec.ulaval.ca (C.G.); mariana.baz@crchudequebec.ulaval.ca (M.B.); sylvie.trottier@crchudequebec.ulaval.ca (S.T.); 2Axe Maladies Infectieuses et Immunitaires, Centre de Recherche du CHU de Québec-Université Laval, Québec, QC G1V 4G2, Canada; 3Biological Risks Department, Institut National de Santé Publique du Québec, Québec, QC G1V 5B3, Canada; nicholas.brousseau@inspq.qc.ca; 4SR Scientific Writing Services, Brossard, QC J4Z 3C7, Canada; 5Department of Chemistry, Université de Montréal, Montréal, QC H2V 0B3, Canada; joelle.pelletier@umontreal.ca (J.N.P.); jf.masson@umontreal.ca (J.-F.M.); 6Department of Biochemistry, Université de Montréal, Montréal, QC H2V 0B3, Canada; 7PROTEO—The Quebec Network for Research on Protein Function, Engineering, and Applications, Québec, QC G1V 0A6, Canada; 8Département de Microbiologie-Infectiologie et d’Immunologie, Faculté de Médecine, Université Laval, Québec, QC G1V 0A6, Canada; 9Quebec Center for Advanced Materials, Université de Montréal, Montréal, QC H3C 3J7, Canada; 10Regroupement Québécois sur les Matériaux de Pointe, Université de Montréal, Montréal, QC H3C 3J7, Canada; 11Centre Interdisciplinaire de Recherche sur le Cerveau et l’Apprentissage, Université de Montréal, Montréal, QC H3C 3J7, Canada; 12Institut Courtois, Université de Montréal, Montréal, QC H2V 0B3, Canada; 13Département de Chimie et Centre d’Optique, Photonique et Laser (COPL), Université Laval, Québec, QC G1V 0A6, Canada; denis.boudreau@chm.ulaval.ca

**Keywords:** SARS-CoV-2, retail workers, cohort profile, COVID-19, influenza vaccine, COVID-19 vaccine

## Abstract

*Background/Objectives:* Retail workers may have been at an increased risk of contracting SARS-CoV-2 during the COVID-19 pandemic. To better understand this group, we set up a longitudinal cohort to document the occurrence of SARS-CoV-2 infection, vaccination uptake and to study immune response. *Methods:* Participants were enrolled between 20 April and 22 October 2021 and attended up to 5 visits over 48 weeks. Information collected was: participant characteristics, SARS-CoV-2 detection tests performed, COVID-19 symptoms, and vaccination (influenza and SARS-CoV-2). *Findings:* We included 304 participants aged 18 to 75; of those, 117 had a first positive SARS-CoV-2 test, mostly (85.5%) during Omicron wave. Forty-two (13.8%) participants got seasonal influenza vaccine within the year (2020–2021) prior to the first visit, and 95.9% had received the primary series of 2 doses of SARS-CoV-2 vaccine by the beginning of Omicron wave. Participants vaccinated for influenza (adjusted hazard ratio (aHR) 2.48; 95% confidence interval (CI): 1.54–3.98) and older patients (aHR 2.39; 95% CI: 1.40–4.10), were more likely to get a first booster of SARS-CoV-2 vaccine compared to those who did not receive influenza vaccine. In contrast, participants who traveled (aHR 0,62; 95% CI: 0.43–0.91) or participated in frequent gatherings (aHR 0.58; 95% CI: 0.39–0.85) were less likely to be boosted. *Conclusions:* Variations in vaccine uptake that are usually observed within populations had little effect on completion of the primary SARS-CoV-2 vaccine series. However, these differences became apparent for booster doses, at a period during which most infections in this cohort were recorded.

## 1. Introduction

During the SARS-CoV-2 pandemic, workers with client-facing duties were considered to be at greater risk of infection than those who worked remotely [1,2,3]. Although most studies have focused on healthcare workers (HCW) [4,5], many non-HCWs were also considered to be at risk due to their occupational exposure [6,7].

Workers in the food and retail industry are an understudied occupational group that may have been at greater risk of contracting SARS-CoV-2. These workers often have below-average incomes, face precarious employment conditions and lack benefit packages to cover health-related absenteeism. At the beginning of the pandemic, these workers often had insufficient training and access to protective equipment to reduce exposure, compared to HCWs [8]. However, risk is likely to vary from one sector to another. For example, grocery and hardware stores were considered essential services and therefore remained open throughout the pandemic, with public health measures (e.g., mask wearing) being enforced and generally well respected. In contrast, restaurants and bars were intermittently opened and closed by health authorities over the same period, and public health measures were more difficult to enforce due to the intrinsically social nature of these businesses and their main purpose-the consumption of food and drink-that precluded continuous mask wearing.

Early evidence confirms that the risk of occupational exposure was high for these workers. In a serological survey conducted in New York City prior to the approval of the first COVID-19 vaccine, the seroprevalence of anti-spike antibodies was higher among grocery store and restaurant workers than in most subgroups of HCWs [1]. In another serosurvey conducted in Switzerland, kitchen staff and grocery store workers exhibited an above average seroprevalence compared to other essential workers [9]. In the Netherlands, individuals working in the hospitality sector were more likely to have a positive PCR test result than those working in non-close-contact occupations [10]. In Japan, restaurants and bars were the second most common setting of COVID-19 outbreaks after healthcare facilities [7,11].

SARS-CoV-2 vaccine was developed and available to the general population approximately one year after the beginning of the COVID-19 pandemic. Vaccines are critical tools to reduce influenza and SARS-CoV-2 morbidity and mortality, but vaccination coverage remain less than optimal among target populations [12]. For instance, in Canada during the 2023–2024 season, influenza vaccine coverage was estimated at 73% among adults aged 65 years and older, and was at 44% among adults aged 18–64 years with chronic medical conditions, both below the national target of 80% [13,14].

Canadian data on high-risk occupational groups remain limited. To date, no thorough investigation of SARS-CoV-2 exposure and vaccine uptake (influenza and SARS-CoV-2) has been conducted among Canadian workers in grocery stores, hardware stores, bars or restaurants [15,16]. Accordingly, we set up a longitudinal cohort that investigated the rate of COVID-19 during the first years of the pandemic, vaccination and the immune response to SARS-CoV-2 in these workers. This article describes the experimental design of the project, the cohort of participants, the occurrence of symptomatic COVID-19 and the uptake of vaccines (influenza and SARS-CoV-2).

## 2. Study Design and Methods

This project was designed early in the COVID-19 pandemic, before the first SARS-CoV-2 vaccine was approved, as a prospective cohort study to investigate SARS-CoV-2 infection and immune response. At that time, the estimated infection rate in our target population was approximately 2–6%, based on limited data from comparable populations (e.g., 1% in Swiss hospital workers and 2.2% in administrative employees versus 29.7% in emergency room workers in Denmark [17,18]. Accordingly, a target enrolment of 600 participants was estimated in order to obtain at least 12 post-COVID-19 blood samples to study immune response. Enrolment was halted at 304 participants as we exceeded our goal of 12 COVID-19 infections, and we chose to reorient our resources and extended the coverage of the study with two additional visits with the already recruited participants.

### 2.1. Participants and Setting

Eligibility criteria included the following: (1) providing informed consent; (2) age ≥18 years; (3) working either on a full-time or part-time basis in a grocery store, hardware store, bar or restaurant located in the administrative regions of Capitale-Nationale and Chaudière-Appalaches that include and surround Québec City, Canada; (4) having a public-facing role in daily work-related activities; (5) having worked ≥20 days between February 1st, 2020 and the first visit; and (6) having no history of hospitalization due to COVID-19.

Participants were recruited using a variety of strategies: (1) an online recruitment campaign conducted by a student-run communication agency; (2) email invitations to members of partner union organizations-Confédération des syndicats nationaux (CSN) and to sectoral organizations of hardware store workers-Association Québécoise de la quincaillerie et des matériaux de construction (AQMAT); and (3) email information to all students and employees at Université Laval and at the Centre Hospitalier Universitaire de Québec in order to publicize the study.

### 2.2. Procedures

The study was initially designed as a prospective cohort with three sampling visits, each separated by 12 ± 2 weeks. In response to the emergence of Omicron, an extension of two visits was proposed to the participants who were still eligible for recruitment at the time of ethics approval. An additional COVID-19 visit (VCoV) was also planned shortly after the occurrence of any SARS-CoV-2 infection during the study period.

At the first visit (“V1”), participants signed an informed consent and were then interviewed by trained nurses to obtain information on demographic, socioeconomic, behavioral, occupational and clinical variables (Appendix A). The questionnaires and selected covariates were adapted from those recommended by our funding body, the COVID-19 Immunity Task Force (CITF), and were based on risk factors for COVID-19 morbidity and mortality established in early epidemiological studies [19]. At or after the third visit (“V3”), eligible participants received information about the extension of the study (two more visits) and signed a new informed consent if they were interested in participating.

At the second (“V2”), third (“V3”), fourth (“V4”) and fifth visits (“V5”), participants completed an abridged version of the V1 questionnaire that focused on SARS-CoV-2 vaccines received, COVID-19 symptoms, SARS-CoV-2 detection (PCR or rapid antigen) tests, exposure and associated risk factors. Blood was drawn and stored to study humoral immunity (i.e., at V1 to V5) and cellular immunity (i.e., at V1, V3, and V5) to SARS-CoV-2. Additional PCR tests were carried out at V4 and V5 to detect asymptomatic carriers.

### 2.3. Study Exposures and Follow-Up

The two main exposures of the study were SARS-CoV-2 infection, defined as a positive virus detection test for SARS-CoV-2 (PCR or antigen detection), and SARS-CoV-2 vaccination. At V1, participants were asked about influenza vaccination in the last season (2020–2021), on any possible or confirmed SARS-CoV-2 infection (i.e., symptoms, virus detection test, date and result) and about their SARS-CoV-2 vaccination history (i.e., number of doses, date of vaccination, type of vaccine) since the beginning of the pandemic, and since the last visit at V2 to V5. Positive SARS-CoV-2 detection tests and vaccination were therefore captured from the beginning of the pandemic until the earliest among the last visit, withdrawal from the study, or loss of eligibility.

### 2.4. Study Outcome

The primary outcomes of the original study were vaccine- and infection-induced immunity. We present here symptomatic infection occurrence and vaccination uptake in this cohort.

### 2.5. Statistical Analysis

Kaplan–Meier curves were used to illustrate the cumulative incidence of vaccination over time. To identify factors associated with vaccination uptake, Cox proportional hazards models were used to estimate adjusted hazard ratios. Variables associated with vaccination in univariate analyses were subsequently included in the multivariate models. All statistical analyses were conducted using RStudio (v2024.12.1 + 563, Posit PBC, Boston, MA, USA). Statistical significance was defined as *p* < 0.05, and all tests were two-tailed.

### 2.6. Ethics Statement

This study was conducted in accordance with the ethical principles outlined in the Belmont Report and the Declaration of Helsinki. All participants provided written informed consent prior to inclusion in the study. This study was approved by the « Comité d’éthique de la recherche du CHU de Québec–Université Laval » (registration number 2021-5744).

### 2.7. Confidentiality and Data Storage

A unique, anonymized identifier was assigned to each participant and used for the data and the blood samples. The samples are stored for up to 10 years, and the data for at least 15 years.

### 2.8. Participants’ Information

More detailed, participant-level information is publicly available on an online platform developed by Maelstrom Research [20]. Data on participants’ immune responses to SARS-CoV-2 infection and vaccination are shared through peer-reviewed publications.

### 2.9. Patient and Public Involvement Statement

No public stakeholders were involved in establishing and designing this cohort.

## 3. Results

### 3.1. Participant Characteristics

The study enrolled 304 individuals between 20 April 2021 and 22 October 2021. The first three visits (V1, V2 and V3; 12 ± 2 weeks) spanned from 20 April 2021 to 9 May 2022. (Figure 1). With the emergence of Omicron, 198 participants who were still within the recruiting window at the time of ethics approval, were included in the extension phase for two additional visits (V4 and V5; 12 ± 4 weeks) between 15 March 2022 and 3 October 2022. Only 13 out of 304 (4.3%) withdrew before V3, and 4 more out of 198 (2.0%) at V5, resulting in a series of at least 5 blood samples drawn over 48 weeks for most participants.

The participants were vaccinated according to recommendations with the vaccines approved by Canadian health authorities: monovalent Comirnaty (Pfizer-BioNTech, Mainz, Germany), Spikevax (Moderna, Cambridge, MA, USA) and Vaxzevria (AstraZeneca, Cambridge, UK), which each required two doses to complete the primary series. By the end of the study (i.e., last visit between 10 May 2022 and 3 October 2022), 95.9% of the participants were fully vaccinated (at least 2 doses) and nearly 70% had received at least one booster dose.

The cohort included 149 (49.0%) restaurant/bar workers, 112 (36.8%) grocery store workers, and 43 (14.1%) hardware store workers (Table 1). On average, participants were aged 41.3 years in the overall cohort. Specifically, restaurants/bar workers were on average 37.2 years old, grocery store workers 44.2 and hardware store workers 48.2. Participants 60 years of age or older represented 15.5% of the cohort. Female participants represented 57.9%. In this cohort, 96.7% self-identified as White, 1.6% as Asian, 1.0% as Latino American, and 0.7% as Black. Levels of education varied: 39.5% reported having a high school diploma or a vocational certificate, 33.2% college education and 22.7% a university degree.

Participants lived and worked mostly in the Capitale-Nationale administrative region, the remainder being in the Chaudière-Appalaches (Appendix A). Most (i.e., 61.8%) lived alone or with one other person, 23.0% lived with children (<18 years), 15.5% with HCWs and 7.6% with teachers or kindergarten workers. These distributions were similar within each occupational group.

According to body mass index (BMI), 41.1% of the participants had a healthy weight (i.e., BMI = 18.5 to 24.9 kg/m^2^), 27.0% were overweight (BMI = 25.0 to 29.9 kg/m^2^) and 30.6% were obese (BMI ≥ 30 kg/m^2^) (Table 2). Only 1.3% were underweight (BMI < 18.5 kg/m^2^). Cigarette (i.e., tobacco) use was reported by 17.4% of the participants and e-cigarette by 7.9%. About a third (i.e., 31.9%) of participants reported having at least one comorbidity associated with an increased risk of hospitalization or death in patients with SARS-CoV-2 infection. Hardware store workers had more comorbidities (48.8%), probably because they were slightly older. Overall, 17.1% reported usually receiving the annual influenza vaccine and 13.8% in the year prior to the first visit (Season 2020–2021).

Approximately half (i.e., 46.4%) of the participants reported working on average more than 30 h per week (Table 3). Most (88.8%) attended at least one gathering of 10 or more persons during the study period, and 40.8% attended more than 10 such gatherings. The predominant mode of transportation was by car (87.5%), followed by bus (12.8%) and walking (9.2%). Traveling outside the province of Québec was reported by 47.0% of the participants, with 25.7% traveling within Canada, 14.8% to the United States, and 27.6% elsewhere. The distribution of participants in each occupational group was similar for the workplace region, mode of transportation and traveling, but differed for the weekly hours worked and the number of gatherings attended.

Overall, 98.4% of the participants reported wearing a mask at work, indicating excellent adherence to this measure (Appendix A). Other measures at work, such as handwashing (98.4%), social distancing (70.1%) and the use of Plexiglas dividers (77.3%) were also frequent. The use of gloves (6.9%) and face-shields (10.5%), which were not extensively promoted by the public health authorities, was less frequent. Outside work, all participants reported wearing a mask in public (100.0%); most avoided usual salutations (85.2%), practiced social distancing (84.2%) and avoided contact with vulnerable persons (83.6%) and crowded places (76.6%). Most participants reported washing their hands when dirty (96.7%), after using the restroom (97.7%), when arriving at (92.1%) and leaving the workplace (71.7%), before eating (87.8%) and after handling trash (79.6%). In general, adherence to these measures was consistently lower among restaurant and bar workers, possibly because of the nature of their work or their younger age (on average).

### 3.2. SARS-CoV-2 Infection and Related Symptoms

A total of 117 participants reported at least one positive test (PCR or rapid antigen) indicating a first infection (Table 4). Eight participants experienced a second positive test (occurring more than 90 days after a previous positive result) during the study.

Among these 117 participants with a first symptomatic infection, 94.9% reported at least one symptom at the time of testing (Table 5). Each individual symptom was experienced by ≥47.9% of participants, except for diarrhea (13.7%) and loss of smell or taste (22.2%). These distributions were similar within each occupational group. A sub-group of 29 participants had symptoms that made them suspect having COVID-19, but PCR or antigen detection test were negative or not performed. Routine PCR tests conducted on asymptomatic participants during visits 4 and 5 returned positive on seven occasions. Of these, four participants had never tested positive, while three had tested positive more than 90 days earlier.

### 3.3. Vaccination for SARS-CoV-2

We next examined vaccination patterns stratified by participant characteristics. Kaplan–Meier event curves showed that both the second dose (primary series) and the third dose (booster) were each administered over relatively short timeframes (Figure 2). It suggests that each wave of vaccination was marked by a concentrated uptake, with little evidence of lagging or delayed vaccination among specific groups. For the primary two-dose series, uptake appeared fairly uniform across participants, indicating a generally consistent adherence. In contrast, the booster dose revealed more variability, particularly across type of work, age, comorbidities, influenza vaccination, attendance at gatherings, and travel history. To formally assess these patterns and identify factors independently associated with vaccination uptake, we conducted Cox regression analyses.

Crude Cox regression analyses identified several factors associated with vaccination uptake (Table 6). Compared to grocery store workers, individuals employed in hardware stores were significantly more likely to receive both two doses (HR = 1.72) and a booster dose (HR = 1.65). Age was also strongly associated with vaccine uptake: participants aged 60–75 were more likely to receive two doses (HR = 1.69) and were three times more likely to receive a booster (HR = 3.05) compared to younger individuals aged 18–39. In contrast, the 40–59 age group showed a significant increase only in booster uptake (HR = 1.59) compared to younger individuals. Individuals with comorbidities were more likely to be vaccinated, particularly with a booster dose (HR = 1.65). Receiving the seasonal influenza vaccine was also strongly associated with COVID-19 vaccine uptake, with increased likelihood for both two-dose (HR = 1.60) and three-dose (HR = 2.70) coverage. Notably, individuals who reported attending gatherings of more than 10 people were less likely to be vaccinated, especially for the booster dose (HR = 0.46; reference ≤10), suggesting a possible perception of risk or behavioral clustering. Travel was significantly associated with a lower uptake of the booster dose (HR = 0.53). Other variables, including sex, education level, BMI, smoking, region of work, number of working hours, and household size, were not significantly associated with vaccine uptake in these crude models.

Adjusted Cox regression analyses revealed a more limited set of factors significantly associated with SARS-VoV-2 vaccine uptake (Table 7). Compared to grocery store workers, those working in hardware stores were more likely to complete a two-dose regimen (aHR 1.49; 95% CI: 1.02–2.18), though the association with booster uptake was not significant. Age remained a key determinant for booster coverage: participants aged 60–75 were significantly more likely to have received a third dose (aHR 2.39; 95% CI: 1.40–4.10), whereas no age group showed significant differences for two-dose completion. Similarly, individuals who had received the seasonal influenza vaccine were more likely to complete both SARS-CoV-2 primary series (aHR 1.51; 95% CI: 1.04–2.19) and a first booster (aHR 2.48; 95% CI: 1.54–3.98). Social behavior also played a role: attending gatherings of more than 10 people was associated with lower booster uptake (aHR 0.58; 95% CI: 0.39–0.85). Travel was also linked to the decreased likelihood of booster coverage (aHR 0.62; 95% CI: 0.43–0.91). Other factors, including comorbidities, sex, education, BMI, smoking, region of work, work hours, and household size, did not show statistically significant associations with COVID-19 vaccine uptake in the adjusted models.

## 4. Discussion

In this longitudinal study, 304 retail workers were observed every 12 weeks for up to 48 weeks to study the occurrence of symptomatic SARS-CoV-2 infection and vaccination among grocery store, hardware store, bar or restaurant workers who lived and worked within the Québec City metropolitan area. The participants provided information on a wide range of demographic, socioeconomic, behavioral, occupational, and clinical variables. Information on COVID-19 symptoms (when applicable), as well as influenza and SARS-CoV-2 vaccination and on any SARS-CoV-2 positive diagnostic test (i.e., PCR or rapid antigen) from the beginning of the pandemic until the last visit was also collected. The study was initiated at a time when all public health measures were implemented and covered seven waves of COVID-19 infection, including those dominated by the Ancestral strain, as well as Alpha, Delta, and Omicron variants, thus capturing a relatively large number of epidemiological periods and infections. Participants had blood sampled at each visit which were stored for later study of immunity to SARS-CoV-2.

A total of 117 first symptomatic COVID-19 were detected, as determined by reported positive PCR or antigen detection test. Most of these infections occurred (85.5%) between December 2021 and October 2022, consistent with the emergence of the highly contagious Omicron variants. The range and frequency of symptoms were similar within each occupational group and showed a pattern consistent with prior studies.

Local public health authorities do not recommend routine influenza vaccination in the age group of our study participants (<75), unless they have comorbidities that increased their risk for severe disease, complications and hospitalization [22,23,24]. One third (32%) of our study participants reported such comorbidities, but only 14% received influenza vaccine during the season of the study. This is in line with previously reported influenza vaccination data [14]. In the present study, SARS-CoV-2 vaccine coverage was high: by the time Omicron had emerged, nearly 95% of the participants had already received complete primary series of two vaccine doses. Participants as a whole, were more inclined to get SARS-CoV-2 vaccine than influenza vaccine, perhaps because of the perceived higher risk of COVID-19, the public health measures in place, and the large media coverage on the impact of this virus. As expected, participants having received the influenza vaccine and older participants (60–75) were more likely to get vaccinated against SARS-CoV-2, probably because they felt more vulnerable [25]. Prior influenza vaccination may also reflect underlying positive attitudes toward vaccines, trust in health authorities, and habitual health behaviors, which are known predictors of uptake for new vaccines such as COVID-19 [26,27]. Finally, participants who engaged in higher-risk activities during the pandemic, such as travel and gatherings, were just as likely to complete the primary vaccination as those who did not, probably reflecting concern early on. However, they were less likely to maintain vaccination over time, and fewer of them received a booster dose as the pandemic progressed and pandemic fatigue set in. Public health measures should be adapted to these phenomena, with vaccine interventions tailored to maintain coverage in high-risk groups.

## 5. Strengths, and Limitations

Some limitations must be considered when interpreting our results. Per study design, none of the participants had experienced a severe COVID-19 illness that required hospitalization before enrolment. Therefore, the cohort may not be used to study population with severe health outcomes, but it is appropriate for mild COVID-19 illness as experienced by most patients. Another limitation is that the cohort may have been subject to a sampling bias since a person willing to participate in a scientific study may be also less apprehensive about receiving a vaccine. Hence, the study participants may not be representative of the overall population of workers. This is suggested by the 5–7% higher SARS-CoV-2 vaccination in our cohort, comparative to the general population of the province of Québec [28]. The high vaccination coverage also made it impossible to assess the impact of vaccination on the risk of infection, since most participants were already vaccinated at any given time.

Variables were excluded from the adjusted models due to low event counts, limited variability, or unstable estimates. Collinearity among certain covariates may have reduced the precision of individual coefficient estimates. Although Kaplan–Meier survival analysis assumes a relatively constant baseline risk for the event, our study involved temporal variation in vaccination “risk” due to changes in vaccine availability. Data collection for all participants was initiated at the onset of the pandemic, aligning their reporting to a shared reference period and sequence of events. As a result, fluctuations in vaccine supply affected the entire cohort equally. The Cox proportional hazards model is designed to accommodate time-varying effects common to all participants, making the calendar time scale of vaccine uptake an appropriate metric for comparisons across participants [29,30,31].

In addition, participant responses on previous COVID-19 infection, symptoms and test may have been affected by a recall bias, particularly for those whose last infection occurred months before the first visit. The low proportion of racial minorities (i.e., 3.3%) is consistent with the size of the visible minority population in the Québec City metropolitan area (i.e., 4.9% according to census) but limits the use of this cohort to study racial determinants of SARS-CoV-2 infection. Lastly, our study relied on symptoms to identify SARS-CoV-2 infection. This may have underestimated the occurrence of SARS-CoV-2 infection, as infection may occur without symptoms and also because most cases happened during the Omicron wave when access to PCR testing was limited, and most infections were self-reported and detected by less sensitive antigen detection. Further analyses incorporating serologic testing offer a more complete picture of COVID-19 episodes [32].

## 6. Conclusions

In this highly vaccinated cohort, most COVID-19 cases occurred during the Omicron waves after nearly universal completion of the primary vaccine series. We show an early and strong adherence to COVID-19 vaccination, followed by a lower booster uptake as pandemic fatigue set in. Booster uptake correlated with participant characteristics, such as older age, influenza vaccination uptake, less frequent travel, and lower participation in social gatherings. These observations provide useful context for planning vaccination strategies in future waves or emerging viral threats.

## Figures and Tables

**Figure 1 idr-17-00122-f001:**
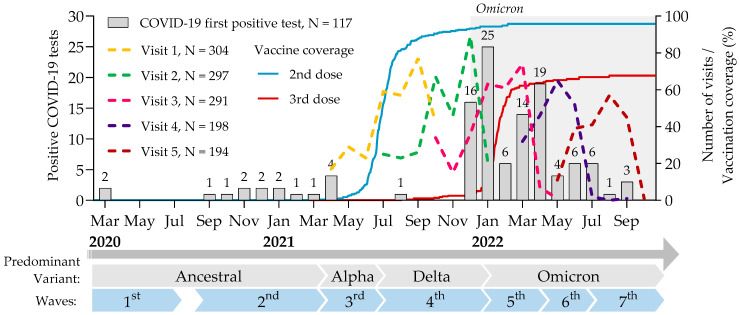
Timeline of the study illustrating the visits (colored dash lines), COVID-19-positive tests (gray bars), and vaccine coverage (colored lines). For vaccination coverage, Kaplan–Meier curves depicted the evolution of the vaccination coverage stratified by the number of doses. Shaded area represents Omicron predominance.

**Figure 2 idr-17-00122-f002:**
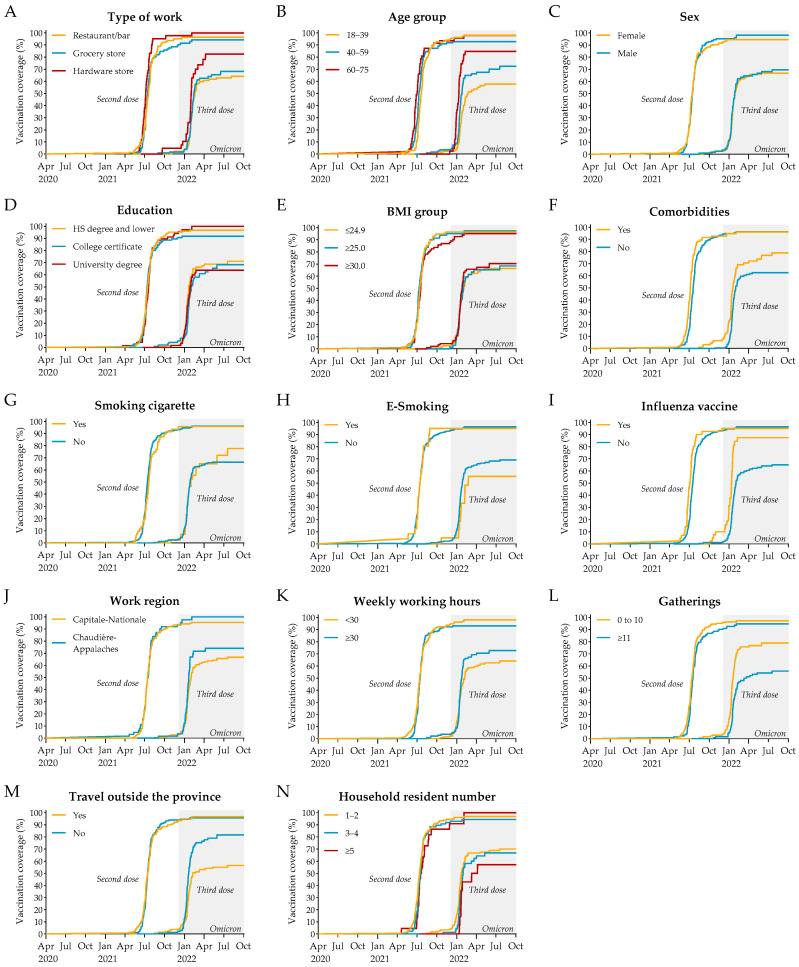
Kaplan–Meier curves depicting the cumulative event rate of complete vaccination (two doses) and booster dose (third dose), stratified by type of work (**A**), age group (**B**), sex (**C**), education level (**D**), BMI group (**E**), comorbidities (**F**), smoking habits (**G**,**H**), influenza vaccine status (**I**), work region (**J**), weekly working hours (**K**), social gatherings frequency (**L**), travel frequency (**M**), and household resident number (**N**). The shaded area indicates the Omicron period.

**Table 1 idr-17-00122-t001:** Detailed demographics of study participants at the first visit.

	Overall Study				Restaurant/Bar				Grocery Store				Hardware Store			
	Population				Workers				Workers				Workers			
	Total		COVID-19 ^1^		Total		COVID-19 ^1^		Total		COVID-19 ^1^		Total		COVID-19 ^1^	
	(N = 304)		(N = 117)		(N = 149)		(N = 62)		(N = 112)		(N = 42)		(N = 43)		(N = 13)	
**Age (years), Mean ± SD**	41.3	±15.9	39.6	±14.6	37.2	±14.8	36.2	±14.3	44.2	±15.3	42.4	±14.1	48.2	±17.3	46.5	±14.6
Age groups, N (%)																
18–59	257	(84.5%)	106	(90.6%)	135	(90.6%)	57	(91.9%)	93	(83.0%)	37	(88.1%)	29	(67.4%)	12	(92.3%)
60–75	47	(15.5%)	11	(9.4%)	14	(9.4%)	5	(8.1%)	19	(17.0%)	5	(11.9%)	14	(32.6%)	1	(7.7%)
**Sex, N (%)**																
Female	176	(57.9%)	72	(61.5%)	95	(63.8%)	39	(62.9%)	57	(50.9%)	25	(59.5%)	24	(55.8%)	8	(61.5%)
Male	128	(42.1%)	45	(38.5%)	54	(36.2%)	23	(37.1%)	55	(49.1%)	17	(40.5%)	19	(44.2%)	5	(38.5%)
**Race/ethnicity, ^2^ N (%)**																
White	294	(96.7%)	114	(97.4%)	142	(95.3%)	61	(98.4%)	109	(97.3%)	40	(95.2%)	43	(100.0%)	13	(100.0%)
Asian	5	(1.6%)	0	(0.0%)	4	(2.7%)	0	(0.0%)	1	(0.9%)	0	(0.0%)	0	(0.0%)	0	(0.0%)
Latino American	3	(1.0%)	1	(0.9%)	3	(2.0%)	1	(1.6%)	0	(0.0%)	0	(0.0%)	0	(0.0%)	0	(0.0%)
Black	2	(0.7%)	2	(1.7%)	0	(0.0%)	0	(0.0%)	2	(1.8%)	2	(4.8%)	0	(0.0%)	0	(0.0%)
**Educational attainment, N (%)**																
Less than high school	14	(4.6%)	3	(2.6%)	4	(2.7%)	2	(3.2%)	9	(8.0%)	0	(0.0%)	1	(2.3%)	1	(7.7%)
High school	79	(26.0%)	27	(23.1%)	42	(28.2%)	14	(22.6%)	23	(20.5%)	7	(16.7%)	14	(32.6%)	6	(46.2%)
Professional	41	(13.5%)	13	(11.1%)	19	(12.8%)	6	(9.7%)	16	(14.3%)	6	(14.3%)	6	(14.0%)	1	(7.7%)
College	101	(33.2%)	45	(38.5%)	51	(34.2%)	26	(41.9%)	35	(31.3%)	14	(33.3%)	15	(34.9%)	5	(38.5%)
University baccalaureate	54	(17.8%)	22	(18.8%)	28	(18.8%)	12	(19.4%)	21	(18.8%)	10	(23.8%)	5	(11.6%)	0	(0.0%)
Graduate studies	15	(4.9%)	7	(6.0%)	5	(3.4%)	2	(3.2%)	8	(7.1%)	5	(11.9%)	2	(4.7%)	0	(0.0%)

^1^ Subset of participants who reported a positive SARS-CoV-2 test (PCR or rapid antigen) at least once during the study period.^2^ Self-reported.

**Table 2 idr-17-00122-t002:** Clinical characteristics of study participants at the first visit.

	Overall Study				Restaurant/Bar				Grocery Store				Hardware Store			
	Population				Workers				Workers				Workers			
	Total		COVID-19 ^1^		Total		COVID-19 ^1^		Total		COVID-19 ^1^		Total		COVID-19 ^1^	
	(N = 304)		(N = 117)		(N = 149)		(N = 62)		(N = 112)		(N = 42)		(N = 43)		(N = 13)	
**BMI scores, Mean ± SD**	27.3	±6.1	27.5	±6.4	27.0	±6.9	26.83	±6.7	28.1	±5.3	27.94	±5.2	26.3	±5.3	27.51	±6.6
<18.5 (underweight)	4	(1.3%)	1	(0.9%)	2	(1.3%)	1	(1.6%)	2	(1.8%)	0	(0.0%)	0	(0.0%)	0	(0.0%)
18.5–24.9 (healthy weight)	125	(41.1%)	49	(41.9%)	73	(49.0%)	30	(48.4%)	29	(25.9%)	13	(31.0%)	23	(53.5%)	6	(46.2%)
25.0–29.9 (overweight)	82	(27.0%)	23	(19.7%)	31	(20.8%)	10	(16.1%)	39	(34.8%)	9	(21.4%)	12	(27.9%)	4	(30.8%)
≥30 (obesity) ^2^	93	(30.6%)	44	(37.6%)	43	(28.9%)	21	(33.9%)	42	(37.5%)	20	(47.6%)	8	(18.6%)	3	(23.1%)
**Smoking, N (%)**																
Cigarette user	53	(17.4%)	22	(18.8%)	33	(22.1%)	14	(22.6%)	16	(14.3%)	6	(14.3%)	4	(9.3%)	2	(15.4%)
E-cigarette user	24	(7.9%)	8	(6.8%)	20	(13.4%)	6	(9.7%)	3	(2.7%)	2	(4.8%)	1	(2.3%)	0	(0.0%)
**Comorbidities ^3^, N (%)**	97	(31.9%)	36	(30.8%)	36	(24.2%)	14	(22.6%)	40	(35.7%)	15	(35.7%)	21	(48.8%)	7	(53.8%)
Arterial hypertension	39	(12.8%)	11	(9.4%)	13	(8.7%)	4	(6.5%)	18	(16.1%)	6	(14.3%)	8	(18.6%)	1	(7.7%)
Lung disease	33	(10.9%)	16	(13.7%)	13	(8.7%)	6	(9.7%)	14	(12.5%)	9	(21.4%)	6	(14.0%)	1	(7.7%)
Diabetes mellitus	18	(5.9%)	5	(4.3%)	4	(2.7%)	1	(1.6%)	11	(9.8%)	3	(7.1%)	3	(7.0%)	1	(7.7%)
Hypothyroidism	17	(5.6%)	4	(3.4%)	6	(4.0%)	2	(3.2%)	6	(5.4%)	1	(2.4%)	5	(11.6%)	1	(7.7%)
Cancer	10	(3.3%)	3	(2.6%)	4	(2.7%)	1	(1.6%)	5	(4.5%)	2	(4.8%)	1	(2.3%)	0	(0.0%)
Cardiovascular disease	8	(2.6%)	2	(1.7%)	2	(1.3%)	0	(0.0%)	3	(2.7%)	1	(2.4%)	3	(7.0%)	1	(7.7%)
Immune deficiency	7	(2.3%)	5	(4.3%)	3	(2.0%)	3	(4.8%)	1	(0.9%)	0	(0.0%)	3	(7.0%)	2	(15.4%)
Chronic neurological disorder	5	(1.6%)	1	(0.9%)	2	(1.3%)	1	(1.6%)	2	(1.8%)	0	(0.0%)	1	(2.3%)	0	(0.0%)
Chronic liver disease	2	(0.7%)	0	(0.0%)	0	(0.0%)	0	(0.0%)	1	(0.9%)	0	(0.0%)	1	(2.3%)	0	(0.0%)
Blood disorder	1	(0.3%)	1	(0.9%)	1	(0.7%)	1	(1.6%)	0	(0.0%)	0	(0.0%)	0	(0.0%)	0	(0.0%)
Obesity ^2^	1	(0%)	0	(0%)	1	(1%)	0	(0%)	0	(0%)	0	(0%)	0	(0%)	0	(0%)
**Influenza vaccination, N (%)**																
Usually received	52	(17.1%)	21	(17.9%)	23	(15.4%)	12	(19.4%)	18	(16.1%)	7	(16.7%)	11	(25.6%)	2	(15.4%)
Season 2020–2021 ^4^	42	(13.8%)	12	(10.3%)	17	(11.4%)	5	(8.1%)	15	(13.4%)	5	(11.9%)	10	(23.3%)	2	(15.4%)

^1^ Subset of participants who reported a positive SARS-CoV-2 test (PCR or rapid antigen) at least once during the study period. ^2^ BMI in the range of obesity was calculated for 93 participants, but only one reported to be obese. ^3^ Comorbidities reported by the participants and associated with an increased risk of hospitalization or death in patients with SARS-CoV-2 infection [21]. ^4^ Influenza vaccine within the year prior to the first visit.

**Table 3 idr-17-00122-t003:** Occupational and behavioral characteristics of study participants.

	Overall Study				Restaurant/Bar				Grocery Store				Hardware Store			
	Population				Workers				Workers				Workers			
	Total		COVID-19 ^1^		Total		COVID-19 ^1^		Total		COVID-19 ^1^		Total		COVID-19 ^1^	
	(N = 304)		(N = 117)		(N = 149)		(N = 62)		(N = 112)		(N = 42)		(N = 43)		(N = 13)	
**Workplace region, ^2^ N (%)**																
Capitale-Nationale	240	(78.9%)	96	(82.1%)	123	(82.6%)	49	(79.0%)	85	(75.9%)	36	(85.7%)	32	(74.4%)	11	(84.6%)
Chaudière-Appalaches	64	(21.1%)	19	(16.2%)	26	(17.4%)	12	(19.4%)	27	(24.1%)	6	(14.3%)	11	(25.6%)	1	(7.7%)
Weekly hours worked, ^3^ Mean ± SD	27.9	±11.5	28.9	±11.4	24.7	±11.0	24.5	±11.0	32.6	±10.4	34.2	±9.8	26.6	±11.9	32.3	±10.4
**Participants working**																
Full time (≥30)	141	(46.4%)	64	(54.7%)	49	(32.9%)	23	(37.1%)	73	(65.2%)	32	(76.2%)	19	(44.2%)	9	(69.2%)
Part time (<30)	163	(53.6%)	53	(45.3%)	100	(67.1%)	39	(62.9%)	39	(34.8%)	10	(23.8%)	24	(55.8%)	4	(30.8%)
**Gathering of 10+ persons, ^3^ Mean ± SD**																
Per group, N (%)																
None	34	(11.2%)	6	(5.1%)	16	(10.7%)	2	(3.2%)	15	(13.4%)	4	(9.5%)	3	(7.0%)	0	(0.0%)
1 to 10 gatherings	146	(48.0%)	47	(40.2%)	57	(38.3%)	16	(25.8%)	62	(55.4%)	22	(52.4%)	27	(62.8%)	9	(69.2%)
11 to 50 gatherings	101	(33.2%)	51	(43.6%)	58	(38.9%)	33	(53.2%)	32	(28.6%)	14	(33.3%)	11	(25.6%)	4	(30.8%)
>50 gatherings	23	(7.6%)	13	(11.1%)	18	(12.1%)	11	(17.7%)	3	(2.7%)	2	(4.8%)	2	(4.7%)	0	(0.0%)
**Transportation, ^2^ N (%)**																
Car	266	(87.5%)	103	(88.0%)	127	(85.2%)	54	(87.1%)	98	(87.5%)	37	(88.1%)	41	(95.3%)	12	(92.3%)
Bus	39	(12.8%)	16	(13.7%)	23	(15.4%)	9	(14.5%)	10	(8.9%)	5	(11.9%)	6	(14.0%)	2	(15.4%)
Walking	28	(9.2%)	11	(9.4%)	13	(8.7%)	6	(9.7%)	14	(12.5%)	5	(11.9%)	1	(2.3%)	0	(0.0%)
Bicycle	13	(4.3%)	6	(5.1%)	7	(4.7%)	3	(4.8%)	4	(3.6%)	2	(4.8%)	2	(4.7%)	1	(7.7%)
Carpooling	2	(0.7%)	2	(1.7%)	1	(0.7%)	1	(1.6%)	1	(0.9%)	1	(2.4%)	0	(0.0%)	0	(0.0%)
**Travel, ^3^ N (%)**																
Any destination	143	(47.0%)	70	(59.8%)	79	(53.0%)	40	(64.5%)	48	(42.9%)	25	(59.5%)	16	(37.2%)	5	(38.5%)
In Canada	78	(25.7%)	39	(33.3%)	45	(30.2%)	24	(38.7%)	23	(20.5%)	13	(31.0%)	10	(23.3%)	2	(15.4%)
To USA	45	(14.8%)	20	(17.1%)	25	(16.8%)	11	(17.7%)	16	(14.3%)	8	(19.0%)	4	(9.3%)	1	(7.7%)
Other destination ^4^	84	(27.6%)	45	(38.5%)	47	(31.5%)	25	(40.3%)	28	(25.0%)	16	(38.1%)	9	(20.9%)	4	(30.8%)

^1^ Subset of participants who reported a positive SARS-CoV-2 test (PCR or rapid antigen) at least once during the study period. ^2^ At the time of the first visit. ^3^ During the entire study period. ^4^ Includes travel to Cuba, Ireland, Great-Britain, Luxembourg, Dominican Republic, South Africa, Bahamas, Morocco, Guadeloupe, Panama, Costa Rica, Greece.

**Table 4 idr-17-00122-t004:** Number of participants with a first SARS-CoV-2 positive test.

	Overall	Bar/	Grocery	Hardware
	Cohort	Restaurant	Store	Store
**Participant reporting**	**117**	**62**	**42**	**13**
PCR	41	25	14	2
Antigen detection	76	37	28	11

Bold: the total.

**Table 5 idr-17-00122-t005:** COVID-19 symptoms at the first SARS-CoV-2 positive test.

	Overall		Bar/	Grocery	Hardware
	Cohort		Restaurant	Store	Store
	(N = 117)		(N = 62)	(N = 42)	(N = 13)
**No symptom, N (%)**	**6**	**(5.1%)**	**2**	**4**	**0**
**Symptoms, N (%)**	**111**	**(94.9%)**	**60**	**38**	**13**
Runny nose or nasal congestion	76	(65.0%)	39	26	11
Cough	74	(63.2%)	41	25	8
Headache	71	(60.7%)	36	26	9
Sore throat	70	(59.8%)	41	23	6
Fever	68	(58.1%)	39	19	9
Muscle pain	65	(55.6%)	35	22	8
Shortness of breath	56	(47.9%)	27	21	8
Loss of sense of smell or taste	26	(22.2%)	11	10	5
Diarrhea	16	(13.7%)	7	8	1

Bold values are the total number of each category.

**Table 6 idr-17-00122-t006:** SARS-CoV-2 Vaccine Uptake, according to influenza vaccination within the year prior to first visit (2020–2021)-Crude analysis.

	Two Doses (Complete Primary Series)		Three Doses (First Booster)	
Characteristic	HR (95% CI)	*p* Value	HR (95% CI)	*p* Value
Type of work				
Grocery Stores	Ref.		Ref.	
Hardware stores	1.72 (1.20–2.47)	0.003	1.65 (1.03–2.64)	0.036
Resto/Bar	1.05 (0.81–1.35)	0.714	0.90 (0.64–1.28)	0.570
Age (years)				
18–39	Ref.		Ref.	
40–59	1.18 (0.92–1.53)	0.201	1.59 (1.11–2.29)	0.012
60–75	1.69 (1.21–2.37)	0.002	3.05 (1.97–4.70)	0.000
Sex				
Female	Ref.		Ref.	
Male	1.04 (0.82–1.31)	0.748	1.06 (0.77–1.46)	0.715
Education				
High school	Ref.		Ref.	
Collge	0.91 (0.70–1.20)	0.518	0.81 (0.57–1.17)	0.272
University	0.96 (0.72–1.29)	0.798	0.85 (0.56–1.28)	0.439
BMI				
<25 (healthy weight)	Ref.		Ref.	
≥25 (overweight)	1.04 (0.78–1.38)	0.796	0.98 (0.65–1.46)	0.907
≥30 (obese)	0.84 (0.63–1.11)	0.213	1.13 (0.78–1.63)	0.519
Comorbidities ^1^				
No	Ref.		Ref.	
Yes	1.50 (1.17–1.93)	0.001	1.65 (1.19–2.29)	0.002
Cigarette smoking				
No	Ref.		Ref.	
Yes	0.86 (0.63–1.17)	0.327	1.10 (0.72–1.67)	0.672
E-Smoking				
No	Ref.		Ref.	
Yes	1.03 (0.66–1.58)	0.913	0.67 (0.33–1.37)	0.275
Influenza vaccine season 2020–2021 ^2^				
No	Ref.		Ref.	
Yes	1.60 (1.14–2.25)	0.006	2.70 (1.79–4.06)	<0.001
Workplace region				
Capitale-Nationale	Ref.		Ref.	
Chaudière-Appalaches	1.09 (0.82–1.44)	0.567	1.32 (0.90–1.94)	0.152
Working hours				
<30 h/w	Ref.		Ref.	
≥30 h/w	0.93 (0.74–1.18)	0.554	1.27 (0.93–1.75)	0.138
Gatherings ^3^				
≤10	Ref.		Ref.	
>10	0.78 (0.61–0.98)	0.037	0.46 (0.33–0.65)	<0.001
Travel				
No	Ref.		Ref.	
Yes ^4^	0.98 (0.77–1.23)	0.833	0.53 (0.38–0.73)	<0.001
Household resident number				
1–2	Ref.		Ref.	
3–4	0.87 (0.68–1.13)	0.302	0.80 (0.56–1.15)	0.236
≥5	0.76 (0.48–1.19)	0.226	0.60 (0.29–1.23)	0.166

HR = crude hazard ratio, CI = confidence interval, Ref. = reference. ^1^ Comorbidities reported by the participants and associated with an increased risk of hospitalization or death in patients with SARS-CoV-2 infection [21], see Table 2 for details. ^2^ Influenza vaccine within the year prior to the first visit. ^3^ Gathering is defined as a meeting of a group of at least ten individuals during the study. ^4^ Travel outside the province of Quebec.

**Table 7 idr-17-00122-t007:** SARS-CoV-2 Vaccination Uptake according to influenza vaccination, within the year prior to first visit (2020–2021)-Adjusted analysis.

	Two Doses (Primary Series)		Three Doses (First Booster)	
Characteristics	aHR (95% CI)	*p* Value	aHR (95% CI)	*p* Value
Type of work				
Grocery Stores	**Ref.**		Ref.	
Hardware stores	**1.49 (1.02–2.18)**	**0.041**	1.21 (0.73–2.02)	0.461
Resto/Bar	1.16 (0.86–1.56)	0.329	1.45 (0.96–2.18)	0.080
Age (years)				
18–39	Ref.		Ref.	
40–59	1.16 (0.85–1.58)	0.338	1.24 (0.81–1.90)	0.312
60–75	1.32 (0.87–20)	0.199	**2.39 (1.40–4.10)**	**0.002**
Sex				
Female	Ref.		Ref.	
Male	1.09 (0.85–1.40)	0.474	1.14 (0.81–1.60)	0.459
Education				
High school	Ref.		Ref.	
College	0.97 (0.73–1.30)	0.855	1.07 (0.73–1.57)	0.728
University	1.03 (0.74–1.45)	0.847	1.31 (0.84–2.05)	0.235
BMI				
<25 (healthy weight)	Ref.		Ref.	
≥25 (overweight)	0.95 (0.69–1.30)	0.742	0.78 (0.5–1.22)	0.271
≥30 (obese)	0.78 (0.57–1.07)	0.124	0.84 (0.55–1.30)	0.435
Comorbidities ^1^				
No	Ref.		Ref.	
Yes	1.3 (0.96–1.77)	0.089	1.1 (0.74–1.63)	0.642
Cigarette smoking				
No	Ref.		Ref.	
Yes	0.89 (0.65–1.24)	0.498	1 (0.63–1.58)	0.994
E-Smoking				
No	Ref.		Ref.	
Yes	1.05 (0.64–1.73)	0.838	0.94 (0.42–2.08)	0.876
Influenza vaccine season 2020–2021 ^2^				
No	Ref.		**Ref.**	
Yes	1.51 (1.04–2.19)	0.030	**2.48 (1.54–3.98)**	**<0.001**
Workplace region				
Capitale-Nationale	Ref.		Ref.	
Chaudière-Appalaches	1.18 (0.87–1.60)	0.279	1.42 (0.93–2.15)	0.100
Working hours				
<30 h/w	Ref.		Ref.	
≥30 h/w	0.98 (0.73–1.32)	0.916	1.25 (0.84–1.85)	0.272
Gatherings ^3^				
≤10	Ref.		**Ref.**	
>10	0.81 (0.61–1.06)	0.129	**0.58 (0.39–0.85)**	**0.006**
Travel				
No	Ref.		**Ref.**	
Yes ^4^	0.97 (0.74–1.27)	0.843	**0.62 (0.43–0.91)**	**0.014**
Household resident number				
1–2	Ref.		Ref.	
3–4	0.99 (0.75–1.31)	0.947	1.09 (0.72–1.64)	0.695
≥5	0.87 (0.52–1.45)	0.584	0.51 (0.23–1.16)	0.111

aHR = adjusted hazard ratio, CI = confidence interval, Ref. = reference. ^1^ Comorbidities reported by the participants and associated with an increased risk of hospitalization or death in patients with SARS-CoV-2 infection [21], see Table 2 for details. ^2^ Influenza vaccine within the year prior to the first visit. ^3^ Gathering is defined as a meeting of a group of at least ten individuals during the study. ^4^ Travel outside the province of Quebec. Bolds help locate statistically significant results.

## Data Availability

More detailed, participant-level information is publicly available on an online platform developed by Maelstrom Research [20]. Researchers with other enquiries or collaboration proposals may contact Sylvie Trottier—the principal investigator in charge of setting up the cohort—at sylvie.trottier@crchudequebec.ulaval.ca.

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
