# Peer review of "Profile, Infection, and Vaccination Uptake: A Cohort of Canadian Retail Workers During the SARS-CoV-2 Pandemic"

_2036-7449, 2025, doi:10.3390/idr17050122_

Round 1

Reviewer 1 Report

Comments and Suggestions for Authors

The study examines a relatively understudied occupational group, Canadian retail workers, during the SARS-CoV-2 pandemic, assessing infection and vaccine uptake. The longitudinal cohort design, with multiple visits and the collection of clinical, occupational, and behavioral data, provides substantial value. However, the presentation and analysis could be strengthened in several areas to enhance clarity, the robustness of conclusions, and the applicability of the findings:

  1. The introduction outlines the occupational risk background but lacks a clear comparative framework with previous Canadian data. Including provincial or national figures on seroprevalence or vaccination coverage, if available, would provide helpful context.
  2. The rationale for including influenza vaccination as a predictor of COVID-19 vaccination should be better justified from both a theoretical and epidemiological perspective.
  3. The definition of a “first episode” of COVID-19 based solely on PCR/antigen testing may have underestimated infections, particularly asymptomatic cases and those occurring before widespread access to rapid tests.
  4. The Cox model for the primary vaccination series yielded very few significant factors; it would be useful to explore potential interactions, such as age × comorbidity or type of work × region.
  5. Potential collinearity between variables (e.g., influenza vaccination and comorbidity, age and type of work) is not discussed.
  6. It is unclear whether the choice of covariates was guided by prior significance criteria or epidemiological theory.
  7. Supplementary tables should present the complete HR and aHR for all variables with 95% CIs.
  8. A sensitivity analysis considering self-reported infections without laboratory confirmation should be included.
  9. The discussion could address implications for occupational health policy, such as targeted campaigns based on risk profile and behavior.
  10. Figures: Some Kaplan–Meier curves are dense and could be simplified or divided into subplots for better readability.

Author Response

The study examines a relatively understudied occupational group, Canadian retail workers, during the SARS-CoV-2 pandemic, assessing infection and vaccine uptake. The longitudinal cohort design, with multiple visits and the collection of clinical, occupational, and behavioral data, provides substantial value. However, the presentation and analysis could be strengthened in several areas to enhance clarity, the robustness of conclusions, and the applicability of the findings:

Comment 1: The introduction outlines the occupational risk background but lacks a clear comparative framework with previous Canadian data. Including provincial or national figures on seroprevalence or vaccination coverage, if available, would provide helpful context.

We thank Reviewer 1 for this review and the constructive comments.

Response 1: To clarify this point, we inserted the following sentence into the introduction:

Line 87: “Canadian data on high-risk occupational groups remain limited.”

Comment 2: The rationale for including influenza vaccination as a predictor of COVID-19 vaccination should be better justified from both a theoretical and epidemiological perspective.

Response 2: Epidemiological Evidence: Several behavioral science models, such as the reasoned action approach and the integrated behavioral model, explain that prior vaccination behavior reflects underlying attitudes, norms, and self-efficacy that influence vaccine decisions. Individuals who get the influenza vaccine tend to have more positive attitudes toward vaccines, greater trust in health authorities, and lower hesitancy, a pattern that generalizes to new vaccines like COVID-19. Theories of habitual health behavior also argue that past action is one of the best predictors of future, similar action, especially when both actions serve the same purpose (preventing infectious disease). PMID: 35705510.

Implications for Research and Public Health: Understanding this relationship helps target interventions: those who routinely get flu shots may be easier to reach with COVID-19 vaccine campaigns, while those who avoid flu vaccines may need tailored messaging to address hesitancy. Including influenza vaccination history as a predictor allows for more precise modeling of vaccine acceptance patterns and enables better segmentation for interventions. PMID: 38461049.

To clarify the rationale for including influenza vaccination as a predictor, we added this information within the Discussion of the revised article:

Lines 379–381: Prior influenza vaccination may also reflect underlying positive attitudes toward vaccines, trust in health authorities, and habitual health behaviors, which are known predictors of uptake for new vaccines such as COVID-19 [26,27].”

Comment 3: The definition of a “first episode” of COVID-19 based solely on PCR/antigen testing may have underestimated infections, particularly asymptomatic cases and those occurring before widespread access to rapid tests.

Response 3: We agree with the reviewer that defining a “first episode” of COVID-19 based solely on PCR or antigen testing may underestimate the true number of infections, particularly in asymptomatic individuals or during periods with limited access to rapid tests. However, this was accounted for in the study design by collecting blood samples for serologic analysis using the official Canadian testing pipeline. For a more comprehensive assessment of COVID-19 episodes, including those identified through serologic testing, we refer readers to the companion publication dedicated to that topic (PMID: 39728020) and added information:

Line 270: “Among these 117 participants with a first symptomatic infection….”

Line 360: “A total of 117 first symptomatic COVID-19 were detected,...”                                      

Lines 414-415: “Further analyses incorporating serologic testing offer a more complete picture of COVID-19 episodes [29].”

Comment 4: The Cox model for the primary vaccination series yielded very few significant factors; it would be useful to explore potential interactions, such as age × comorbidity or type of work × region.

Response 4: The Cox models reported in the article included all covariates simultaneously in the adjusted analysis. Potential interactions were accounted for within this framework.

Comment 5: Potential collinearity between variables (e.g., influenza vaccination and comorbidity, age and type of work) is not discussed.

Response 5: We agree that potential collinearity between variables, such as influenza vaccination and comorbidity or age and type of work, could influence the estimates in the adjusted models. We addressed this point in the Discussion section of the revised article and acknowledged the possibility of residual bias due to collinearity among certain covariates:

Lines 402-403: “Collinearity among certain covariates may have reduced the precision of individual coefficient estimates.”

Comment 6: It is unclear whether the choice of covariates was guided by prior significance criteria or epidemiological theory.

Response 6: The selection of covariates was guided by established epidemiological evidence early in the pandemic. Specifically, the variables included in our models were those identified as risk factors for COVID-19–related morbidity and mortality. These covariates were determined and recommended by the Canadian COVID-19 Immunity Task Force (CITF), our federal funding agency, to ensure consistency and comparability across studies funded under their mandate. As such, the choice of covariates reflects both prior evidence and the need for harmonization across the national research effort. We revised the methodology to provide additional clarification:

Lines 132-135:The questionnaires and selected covariates were adapted from those recommended by our funding body, the COVID-19 Immunity Task Force (CITF), and were based on risk factors for COVID-19 morbidity and mortality established in early epidemiological studies [19].”

Comment 7: Supplementary tables should present the complete HR and aHR for all variables with 95% CIs.

Response 7: We thank the reviewer for this suggestion. We agree that comprehensive reporting of unadjusted and adjusted hazard ratios can be helpful for readers. However, several variables not retained in the final models lacked sufficient sample size to yield stable or interpretable adjusted estimates. Including these would risk introducing noise and obscuring the core findings. We have now acknowledged this limitation in the Discussion section:

Lines 401-403: “Variables were excluded from the adjusted models due to low event counts, limited variability, or unstable estimates.”

Comment 8: A sensitivity analysis considering self-reported infections without laboratory confirmation should be included.

Response 8: We thank the reviewer for this suggestion. While a sensitivity analysis including self-reported infections without laboratory confirmation would be interesting, it is beyond the scope of the current article. This work focuses specifically on the description of the cohort and on the factors that influenced the decision of being vaccinated against COVID-19. This comment was accounted for in the study design by collecting blood samples for serologic analysis using the official Canadian testing pipeline; these results are presented in a separate article (PMID: 39728020). For a more comprehensive assessment of COVID-19 episodes, including those identified through serologic testing, we refer readers to the companion publication dedicated to that topic (PMID: 39728020) and added information:

Lines 409-415: Lastly, our study relied on symptoms to identify SARS-CoV-2 infection. This may have underestimated the occurrence of SARS-CoV-2 infection, as infection may occur without symptoms and also because most cases happened during the Omicron wave when access to PCR testing was limited, and most infections were self-reported and detected by less sensitive antigen detection. Further analyses incorporating serologic testing offer a more complete picture of COVID-19 episodes [29].

Comment 9: The discussion could address implications for occupational health policy, such as targeted campaigns based on risk profile and behavior.

Response 9: We have addressed the comment by adding a sentence to the Discussion highlighting the implications for occupational health policy, including targeted vaccine interventions for high-risk groups and maintenance of routine vaccinations during pandemics. Specifically, we added:

Lines 386-387: “Public health measures should be adapted to these phenomena, with vaccine interventions tailored to maintain coverage in high-risk occupational groups.”

Comment 10: Figures: Some Kaplan–Meier curves are dense and could be simplified or divided into subplots for better readability.

Response 10: Data of the different groups are similar and close together resulting in a dense visualisation. We believe that sub plotting will have little to no impact in improving readability.

Reviewer 2 Report

Comments and Suggestions for Authors

This is a well-written report on a well-designed and executed prospective surveillance study of COVID-19 infection and vaccination in a population whose occupation would be expected to put them at risk for COVID-19 infection. It provides useful information on both pandemic exposure and vaccination uptake in this group.

Minor points:

lines 33-34, it would be helpful to identify the start date in the abstract

lines 82-85, clearer if specify influenza vaccination is what is reported

lines 97-98, 12 volunteers, not clear what this means, out of 600 you only expected 12?

line 99, why stop at 304?

lines 378-379, “This is suggested by the higher (5-7%) primary series uptake compared to the general population of the province of Québec”, is 5-7% correct for uptake? As per lines 189-191, “By the end of the study (i.e., last visit between May 10th, 2022 and October 3rd, 2022), 95.9% of the participants were fully vaccinated (at least 2 doses) and nearly 70% had received at least one booster dose.”

lines 387-390, “Lastly, our study may have underestimated the occurrence of SARS-CoV-2 infection, as most cases happened during the Omicron wave when access to PCR testing was limited, antigen tests were less sensitive, and infections were self-reported rather than identified through laboratory surveillance.”, is this correct?

Author Response

This is a well-written report on a well-designed and executed prospective surveillance study of COVID-19 infection and vaccination in a population whose occupation would be expected to put them at risk for COVID-19 infection. It provides useful information on both pandemic exposure and vaccination uptake in this group.

Minor points:

Comment 1: lines 33-34, it would be helpful to identify the start date in the abstract

We would like to thank Reviewer 2 for the thoughtful assessment of our manuscript.

Response 1: We have included the window of enrolment in the study within the abstract:

Lines 33-34: “Methods: Participants were enrolled between April 20th and October 22nd, 2021 and attended up to 5 visits over 48 weeks.”

Comment 2: lines 82-85, clearer if specify influenza vaccination is what is reported

Response 2: We have specified the nature of the coverage with the text:

Line 83-86: “For instance, in Canada during the 2023–2024 season, influenza vaccine coverage was estimated at 73% among adults aged 65 years and older, and was at 44% among adults aged 18–64 years with chronic medical conditions, both below the national target of 80%[13,14].

Comment 3: lines 97-98, 12 volunteers, not clear what this means, out of 600 you only expected 12?

Response 3: Yes. This project was designed early in the COVID-19 pandemic and limited seroprevalence data from populations similar to ours showed 1% in Swiss hospital workers (PMID: 33028454) and 2.2% in administrative employees versus 29.7% in emergency room workers in Denmark (PMID: 33011792). Based on these data, we estimated an infection rate of 2–6% in our study population, which corresponds to at least 12 infections among the 600 participants recruited. We have added clarification in the Study Design section to provide context on expected infection rates and recruitment:

Lines 98–105:At that time, the estimated infection rate in our target population was approximately 2–6%, based on limited data from comparable populations (e.g., 1% in Swiss hospital workers and 2.2% in administrative employees versus 29.7% in emergency room workers in Denmark [17,18]. Accordingly, a target enrolment of 600 participants was estimated in order to obtain at least 12 post-COVID-19 blood samples to study immune response.

Comment 4: line 99, why stop at 304?

Response 4: We halted the recruitment at 304 because we exceeded our goal of at least 12 COVID-19 cases; we already had 17 COVID-19 cases at the enrolment of the 300th participant and 2 more visits to go. With the emergence of Omicron, we chose to reorient our resources and extend the coverage of the study with two additional visits with the already recruited participants:

Lines 105–108.Enrolment was halted at 304 participants as we exceeded our goal of 12 COVID-19 infections and we chose to reorient our resources and extend the coverage of the study with two additional visits with the already recruited participants.

Comment 5: lines 378-379, “This is suggested by the higher (5-7%) primary series uptake compared to the general population of the province of Québec”, is 5-7% correct for uptake? As per lines 189-191, “By the end of the study (i.e., last visit between May 10th, 2022 and October 3rd, 2022), 95.9% of the participants were fully vaccinated (at least 2 doses) and nearly 70% had received at least one booster dose.”

Reponse 5: We clarified the sentence as follows:

Lines 396-398: “This is suggested by the 5-7% higher SARS-CoV-2 vaccination in our cohort, comparative the general population of the province of Québec [28].

Comment 6: lines 387-390, “Lastly, our study may have underestimated the occurrence of SARS-CoV-2 infection, as most cases happened during the Omicron wave when access to PCR testing was limited, antigen tests were less sensitive, and infections were self-reported rather than identified through laboratory surveillance.”, is this correct?

Response 6: We clarified this as follows:

Line 409-415: Lastly, our study relied on symptoms to identify SARS-CoV-2 infection. This may have underestimated the occurrence of SARS-CoV-2 infection, as infection may occur without symptoms and also because most cases happened during the Omicron wave when access to PCR testing was limited, and most infections were self-reported and detected by less sensitive antigen detection. Further analyses incorporating serologic testing offer a more complete picture of COVID-19 episodes [29].”

Reviewer 3 Report

Comments and Suggestions for Authors

The manuscript is mainly a description of a study focused on retail workers in Quebec, Canada.  My comments are listed below:

  1. There are too many descriptive statistics of the sample that most readers would not feel interested in. For example, the sample demographics and characteristics presented in Tables 1 through 3 are simple numbers and percentages of the same participants. Information in Table 4 can be expressed in one sentence. The space of journal articles is previous. It is questionable to have 7 tables in a journal article.
  2. The only analytical information in the article is the Kaplan-Meier survival analysis. Usually, in survival analysis, the dependent variable (DV) is the exposure time to the event, which means the event has a critical importance in the analysis. I am not sure if "vaccination" is an appropriate event. During the COVID-19 pandemic, vaccination was highly dependent on the availability of the vaccines. The relationship between the passage of time and vaccination is weak. 
  3. It seems to me that the manuscript is just the descriptive part of the project. It does not reveal much of some other important findings. 

Author Response

The manuscript is mainly a description of a study focused on retail workers in Quebec, Canada.  My comments are listed below:

Comment 1: There are too many descriptive statistics of the sample that most readers would not feel interested in. For example, the sample demographics and characteristics presented in Tables 1 through 3 are simple numbers and percentages of the same participants. Information in Table 4 can be expressed in one sentence. The space of journal articles is previous. It is questionable to have 7 tables in a journal article.

We thank Reviewer 3 for their careful review of our manuscript.

Response 1: We believe it is important to share this information for transparency and to help better understand the outcomes of the study. As we refer to these tables within the text and the analysis, so we believe that it is easier for the reader to have them readily available.

Comment 2: The only analytical information in the article is the Kaplan-Meier survival analysis. Usually, in survival analysis, the dependent variable (DV) is the exposure time to the event, which means the event has a critical importance in the analysis. I am not sure if "vaccination" is an appropriate event. During the COVID-19 pandemic, vaccination was highly dependent on the availability of the vaccines. The relationship between the passage of time and vaccination is weak. 

Response 2: Although Kaplan–Meier survival analysis assumes a roughly constant baseline risk of the event, in our study the “risk” of vaccination varied over time due to vaccine availability. However, all participants were affected equally. Consequently, the relative timing of vaccine uptake remains a valid measure for assessing disparities in vaccine acceptance and access, allowing meaningful comparisons across participants.

Comment 3: It seems to me that the manuscript is just the descriptive part of the project. It does not reveal much of some other important findings. 

Response 3: This manuscript is focused on describing the cohort and exploring COVID-19 vaccine uptake across participant characteristics. Our aim is to provide a detailed epidemiological profile of a specific essential workforce. While the article may appear primarily descriptive, we believe it contributes valuable insights for public health planning, particularly in identifying groups with lower vaccine uptake, and helps to pinpoint some of the factors influencing decision of vaccination (lines 369-371). These findings can help guide more targeted vaccination efforts in future public health emergencies. We recognize that the relevance of such findings may vary depending on the reader’s disciplinary background, but we believe they are of importance to public health practitioners and policymakers (lines 374-380).

Lines 369-371: As expected, participants having received the influenza vaccine and older participants (60-75) were more likely to get vaccinated against SARS-CoV-2, probably because they felt more vulnerable [25].

Lines 374-380: Finally, participants who engaged in higher-risk activities during the pandemic, such as travel and gatherings, were just as likely to complete the primary vaccination as those who did not, probably reflecting concern early on. However, they were less likely to maintain vaccination over time, and fewer of them received a booster dose as the pandemic progressed and pandemic fatigue set in. Public health measures should be adapted to these phenomena, with vaccine interventions tailored to maintain coverage in high-risk groups.

Round 2

Reviewer 1 Report

Comments and Suggestions for Authors

The authors have resolved my doubts and improved the paper.

Author Response

Comment 1: The authors have resolved my doubts and improved the paper.

Response 1: We thank the reviewer for their thorough evaluation and constructive feedback. We appreciate the time and effort they dedicated to reviewing our manuscript, which has helped us improve the quality of our work.